# Multi-View Attention Network to Improve Breast Cancer Detection

**Wen Tai**[*1]                   WENTAI@MICB2B.COM

[1] *Marketech International Corp, Taipei, Taiwan*

**Pin-Jui Huang**[*1]             JEFFERYHUANG@MICB2B.COM

**Dongmyung Shin**[†2]          SHINSAE11@RADISENTECH.COM

[2] *Radisen Co. Ltd., Seoul, South Korea*

**Editors:** Accepted for publication at MIDL 2024

## Abstract

Breast cancer is the most prevalent cancer in women, and mammography is an effective imaging modality for detecting it in its early stages. However, identifying tumors in mammograms is challenging, and many AI algorithms have been proposed to assist radiologists in detecting them. This study focuses on demonstrating the potential of a multi-view attention network for breast cancer detection by investigating the change in the detection performance depending on the types of attention (no, single-view, or multi-view attention), image resolution (low or high), and backbone network (ResNet50 or HRNet). The experiment results showed that the detection performance of a high-resolution, multi-view attention network with an HRNet backbone was better than the other networks with different configurations, suggesting that multi-view attention has benefits in detecting masses on mammograms.

**Keywords:** Object Detection, Attention, Breast Cancer, Mass Detection, Mammogram

## 1. Introduction

Breast cancer is one of the most prevalent cancers worldwide and reports the highest mortality rate in women (Ferlay et al., 2019). Mammography has proven to be an effective imaging modality to detect breast cancer in its early stages (Smith et al., 2019). However, identifying tumors in mammograms is challenging due to their various sizes and shapes (Razzak et al., 2018), breast densities among patients (Boyd et al., 2007), and radiologists' fatigue and levels of expertise (Fenton et al., 2007). Therefore, many AI algorithms have been proposed to assist radiologists in detecting tumors on mammograms by providing their locations (Ribli et al., 2018). Some recent works have demonstrated the benefits of AI assistance, such as reducing turnaround time (Rodriguez-Ruiz et al., 2019) and increasing detection rate (McKinney et al., 2020).

In recent years, an attention mechanism has been gaining popularity, resulting in state-of-the-art performance in the field of object detection (Hu et al., 2018). The attention mechanism, in its nature, shares similarities with how radiologists review mammograms, where radiologists compare multiple views of the same patient to confirm the presence of tumors. Based on this observation, a few studies have proposed methods that applied the attention mechanism for breast cancer detection (Ma et al., 2021; Yang et al., 2021; Truong Vu

---

[*] Contributed equally

[†] Corresponding author

et al., 2023). However, none of them have investigated the effect of the number of views, input image resolution, and backbone network on the performance of an attention-based model. This study aims to validate AI models with different configurations to demonstrate the potential of a multi-view attention network for breast cancer detection.

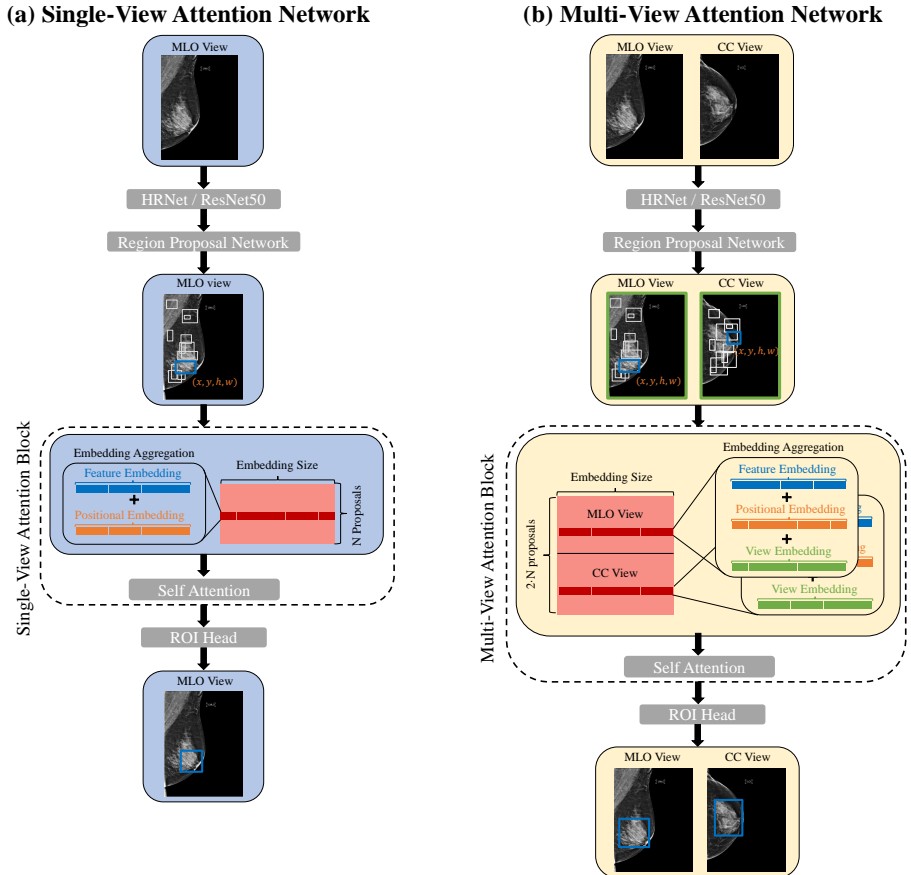

Figure 1: Proposed Faster RCNN with (a) single-view and (b) multi-view attention blocks.

## 2. Method

**Attention Network** Figure 1 shows the network architectures with the single-view and multi-view attention blocks. Based on a faster RCNN network (Ren et al., 2015), we added a self-attention layer (Vaswani et al., 2017) between a region proposal network (RPN) and an ROI head. In the single-view attention network, we only added positional embedding (Vaswani et al., 2017) to feature embedding of each proposal (see Single-View Attention Block in Figure 1a). On the other hand, in the multi-view attention network, we also added view embedding to feature embedding, along with positional embedding (see Multi-View Attention Block in Figure 1b).

**Experiments** We trained and tested AI models using a digital database for screening mammography dataset (80% for training; 20% for testing) (Lee et al., 2017), considering

mass annotations only. We conducted repetitive experiments by changing the models' configuration as follows: low and high resolution of input images (1024 by 768 vs. 2100 by 1700), two backbone networks' architectures (ResNet50 (He et al., 2016) vs. HRNet (Wang et al., 2020)), and three attention processes (w/o/ attention vs. single-view attention vs. multi-view attention). We evaluated the detection performance of each AI model using different combinations of these configurations by calculating recalls at false positives per image (FPPI) of 0.5, 1.0, and 2.0.

**Hyperparameters** We used the faster RCNN network's parameter settings described in (Ribli et al., 2018), with the exception of the number of boxes in the ROI head during training = 32 and positive ratio = 0.33. For the experiments with high-resolution input images (i.e., 2100 by 1700), we set the anchor sizes of an RPN to 64, 128, 256, 480, and 512 to match the scaled sizes of tumors in high-resolution. We trained the model for a maximum of 20 epochs and selected the best model that achieved the highest average recall over different FPPIs. The ResNet50 and HRNet used as backbone networks were pre-trained using ImageNet.

Table 1: Detection performance of AI models in localizing masses on mammograms with different input image resolution, backbone networks, and attention processes.

| Resolution | Backbone | Attention | Recall@0.5 | Recall@1.0 | Recall@2.0 |
|---|---|---|---|---|---|
| 1024 x 768 | ResNet50 | w/o/ attention | 0.672 | 0.754 | 0.823 |
| 1024 x 768 | ResNet50 | single-view | 0.66 | 0.758 | 0.831 |
| 1024 x 768 | ResNet50 | multi-view | 0.681 | 0.762 | 0.834 |
| 1024 x 768 | HRNet | w/o/ attention | 0.716 | 0.784 | 0.841 |
| 1024 x 768 | HRNet | single-view | 0.712 | 0.778 | 0.834 |
| 1024 x 768 | HRNet | multi-view | 0.701 | 0.793 | 0.843 |
| 2100 x 1700 | HRNet | w/o/ attention | 0.729 | 0.796 | 0.864 |
| 2100 x 1700 | HRNet | single-view | 0.741 | 0.804 | **0.867** |
| 2100 x 1700 | HRNet | multi-view | **0.769** | **0.832** | 0.866 |

## 3. Result

Table 1 summarizes the experimental results. In the same resolution, the detection performance of the attention networks with an HRNet backbone was consistently better than those with a ResNet50 backbone, indicating the superiority of an HRNet in extracting features for the RPN (Yang et al., 2021). Compared to the low-resolution experiments, the high-resolution experiments reported higher recalls over all FPPIs, implying that the high resolution is crucial in identifying masses correctly. Applying multi-view attention produced better results than single-view attention or no attention. This suggests that multi-view attention has benefits in detecting masses on mammograms by utilizing the features from multiple views of the same patient concurrently, similar to how radiologists do.

## Acknowledgments

We thank all AI team members in Marketech lnternational Corp. and Radisen Co. Ltd.

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
