# OpenReview forum: "Multi-View Attention Network to Improve Breast Cancer Detection"
_MIDL.io/2024/Short_Papers — MIDL 2024 Short Papers_

### Official Review · Reviewer_1JQm · 2024-04-17

**Confidence:** 5
**Final Rating:** 3.5

**Review:**

This study explores the potential of a multi-view attention network for breast cancer detection by examining how detection performance varies with different attention types (no, single-view, or multi-view attention), image resolutions (low or high), and backbone networks. Similar works have been investigated before and there is no technical novelty. However, the investigation of different views/resolutions/backbone networks makes the work an interesting application paper.

---

### Decision · Program_Chairs · 2024-04-26

Accept